# Score-based Data Assimilation

**François Rozet**
University of Liège
francois.rozet@uliege.be

**Gilles Louppe**
University of Liège
g.louppe@uliege.be

## Abstract

Data assimilation, in its most comprehensive form, addresses the Bayesian inverse problem of identifying plausible state trajectories that explain noisy or incomplete observations of stochastic dynamical systems. Various approaches have been proposed to solve this problem, including particle-based and variational methods. However, most algorithms depend on the transition dynamics for inference, which becomes intractable for long time horizons or for high-dimensional systems with complex dynamics, such as oceans or atmospheres. In this work, we introduce score-based data assimilation for trajectory inference. We learn a score-based generative model of state trajectories based on the key insight that the score of an arbitrarily long trajectory can be decomposed into a series of scores over short segments. After training, inference is carried out using the score model, in a non-autoregressive manner by generating all states simultaneously. Quite distinctively, we decouple the observation model from the training procedure and use it only at inference to guide the generative process, which enables a wide range of zero-shot observation scenarios. We present theoretical and empirical evidence supporting the effectiveness of our method.

## 1 Introduction

Data assimilation (DA) [1–9] is at the core of many scientific domains concerned with the study of complex dynamical systems such as atmospheres, oceans or climates. The purpose of DA is to infer the state of a system evolving over time based on various sources of imperfect information, including sparse, intermittent, and noisy observations.

Formally, let $x_{1:L} = (x_1, x_2, \ldots, x_L) \in \mathbb{R}^{L \times D}$ denote a trajectory of states in a discrete-time stochastic dynamical system and $p(x_{i+1} \mid x_i)$ be the transition dynamics from state $x_i$ to state $x_{i+1}$. An observation $y \in \mathbb{R}^M$ of the state trajectory $x_{1:L}$ follows an observation process $p(y \mid x_{1:L})$, generally formulated as $y = \mathcal{A}(x_{1:L}) + \eta$, where the measurement function $\mathcal{A} : \mathbb{R}^{L \times D} \mapsto \mathbb{R}^M$ is often non-linear and the observational error $\eta \in \mathbb{R}^M$ is a stochastic additive term that accounts for instrumental noise and systematic uncertainties. In this framework, the goal of DA is to solve the inverse problem of inferring plausible trajectories $x_{1:L}$ given an observation $y$, that is, to estimate the trajectory posterior

$$p(x_{1:L} \mid y) = \frac{p(y \mid x_{1:L})}{p(y)} p(x_1) \prod_{i=1}^{L-1} p(x_{i+1} \mid x_i) \tag{1}$$

where the initial state prior $p(x_1)$ is commonly referred to as background [5–9]. In geosciences, the amount of data available is generally insufficient to recover the full state of the system from the observation alone [8]. For this reason, the physical model underlying the transition dynamics is of paramount importance to fill in spatial and temporal gaps in the observation.

State-of-the-art approaches to data assimilation are based on variational assimilation [1, 2, 5–7]. Many of these approaches formulate the task as a maximum-a-posteriori (MAP) estimation problem and solve it by maximizing the log-posterior density $\log p(x_{1:L} \mid y)$ via gradient ascent. Although this

approach only produces a point estimate of the trajectory posterior, its cost can already be substantial for problems of the size and complexity of geophysical systems, since it requires differentiating through the physical model. The amount of data that can be assimilated is therefore restricted because of computational limitations. For example, only a small volume of the available satellite data is exploited for operational forecasts and yet, even with these restrictions, data assimilation accounts for a significant fraction of the computational cost for modern numerical weather prediction [10, 11]. Recent work has shown that deep learning can be used in a variety of ways to improve the computational efficiency of data assimilation, increase the reconstruction performance by estimating unresolved scales after data assimilation, or integrate multiple sources of observations [12–19].

**Contributions**   In this work, we propose a novel approach to data assimilation based on score-based generative models. Leveraging the Markovian structure of dynamical systems, we train a score network from short segments of trajectories which is then capable of generating physically consistent and arbitrarily-long state trajectories. The observation model is decoupled from the score network and used only during assimilation to guide the generative process, which allows for a wide range of zero-shot observation scenarios. Our approach provides an accurate approximation of the whole trajectory posterior – it is not limited to point estimates – without simulating or differentiating through the physical model. The code for all experiments is made available at https://github.com/francois-rozet/sda.

## 2   Background

Score-based generative models have recently shown remarkable capabilities, powering many of the latest advances in image, video or audio generation [20–27]. In this section, we review score-based generative models and outline how they can be used for solving inverse problems.

**Continuous-time score-based generative models**   Adapting the formulation of Song et al. [28], samples $x \in \mathbb{R}^D$ from a distribution $p(x)$ are progressively perturbed through a continuous-time diffusion process expressed as a linear stochastic differential equation (SDE)

$$\mathrm{d}x(t) = f(t)\,x(t)\,\mathrm{d}t + g(t)\,\mathrm{d}w(t) \tag{2}$$

where $f(t) \in \mathbb{R}$ is the drift coefficient, $g(t) \in \mathbb{R}$ is the diffusion coefficient, $w(t) \in \mathbb{R}^D$ denotes a Wiener process (standard Brownian motion) and $x(t) \in \mathbb{R}^D$ is the perturbed sample at time $t \in [0, 1]$. Because the SDE is linear with respect to $x(t)$, the perturbation kernel from $x$ to $x(t)$ is Gaussian and takes the form

$$p(x(t) \mid x) = \mathcal{N}(x(t) \mid \mu(t)\,x, \Sigma(t)) \tag{3}$$

where $\mu(t)$ and $\Sigma(t) = \sigma(t)^2 I$ can be derived analytically from $f(t)$ and $g(t)$ [29, 30]. Denoting $p(x(t))$ the marginal distribution of $x(t)$, we impose that $\mu(0) = 1$ and $\sigma(0) \ll 1$, such that $p(x(0)) \approx p(x)$, and we chose the coefficients $f(t)$ and $g(t)$ such that the influence of the initial sample $x$ on the final perturbed sample $x(1)$ is negligible with respect to the noise level – that is, $p(x(1)) \approx \mathcal{N}(0, \Sigma(1))$. The variance exploding (VE) and variance preserving (VP) SDEs [28, 31, 32] are widespread examples satisfying these constraints.

Crucially, the time reversal of the forward SDE (2) is given by a reverse SDE [28, 33]

$$\mathrm{d}x(t) = \left[ f(t)\,x(t) - g(t)^2\,\nabla_{x(t)} \log p(x(t)) \right] \mathrm{d}t + g(t)\,\mathrm{d}w(t). \tag{4}$$

That is, we can draw noise samples $x(1) \sim \mathcal{N}(0, \Sigma(1))$ and gradually remove the noise therein to obtain data samples $x(0) \sim p(x(0))$ by simulating the reverse SDE from $t = 1$ to $0$. This requires access to the quantity $\nabla_{x(t)} \log p(x(t))$ known as the score of $p(x(t))$.

**Denoising score matching**   In practice, the score $\nabla_{x(t)} \log p(x(t))$ is approximated by a neural network $s_\phi(x(t), t)$, named the score network, which is trained to solve the denoising score matching objective [28, 34, 35]

$$\arg\min_\phi \mathbb{E}_{p(x)p(t)p(x(t)|x)} \left[ \sigma(t)^2 \left\| s_\phi(x(t), t) - \nabla_{x(t)} \log p(x(t) \mid x) \right\|_2^2 \right] \tag{5}$$

where $p(t) = \mathcal{U}(0, 1)$. The theory of denoising score matching ensures that $s_\phi(x(t), t) \approx \nabla_{x(t)} \log p(x(t))$ for a sufficiently expressive score network. After training, the score network

is plugged into the reverse SDE (4), which is then simulated using an appropriate discretization scheme [28, 30, 36, 37].

In practice, the high variance of $\nabla_{x(t)} \log p(x(t) \mid x)$ near $t = 0$ makes the optimization of (5) unstable [30]. To mitigate this issue, a slightly different parameterization $\epsilon_\phi(x(t), t) = -\sigma(t) s_\phi(x(t), t)$ of the score network is often used, which leads to the otherwise equivalent objective [30, 32, 36]

$$\arg\min_\phi \mathbb{E}_{p(x)p(t)p(\epsilon)} \left[ \|\epsilon_\phi(\mu(t) x + \sigma(t) \epsilon, t) - \epsilon\|_2^2 \right] \tag{6}$$

where $p(\epsilon) = \mathcal{N}(0, I)$. In the following, we keep the score network notation $s_\phi(x(t), t)$ for convenience, even though we adopt the parameterization $\epsilon_\phi(x(t), t)$ and its objective for our experiments.

**Zero-shot inverse problems**   With score-based generative models, we can generate samples from the unconditional distribution $p(x(0)) \approx p(x)$. To solve inverse problems, however, we need to sample from the posterior distribution $p(x \mid y)$. This could be accomplished by training a conditional score network $s_\phi(x(t), t \mid y)$ to approximate the posterior score $\nabla_{x(t)} \log p(x(t) \mid y)$ and plugging it into the reverse SDE (4). However, this would require data pairs $(x, y)$ during training and one would need to retrain a new score network each time the observation process $p(y \mid x)$ changes. Instead, many have observed [28, 38–41] that the posterior score can be decomposed into two terms thanks to Bayes' rule

$$\nabla_{x(t)} \log p(x(t) \mid y) = \nabla_{x(t)} \log p(x(t)) + \nabla_{x(t)} \log p(y \mid x(t)). \tag{7}$$

Since the prior score $\nabla_{x(t)} \log p(x(t))$ can be approximated with a single score network, the remaining task is to estimate the likelihood score $\nabla_{x(t)} \log p(y \mid x(t))$. Assuming a differentiable measurement function $\mathcal{A}$ and a Gaussian observation process $p(y \mid x) = \mathcal{N}(y \mid \mathcal{A}(x), \Sigma_y)$, Chung et al. [41] propose the approximation

$$p(y \mid x(t)) = \int p(y \mid x) \, p(x \mid x(t)) \, \mathrm{d}x \approx \mathcal{N}\left(y \mid \mathcal{A}(\hat{x}(x(t))), \Sigma_y\right) \tag{8}$$

where the mean $\hat{x}(x(t)) = \mathbb{E}_{p(x \mid x(t))}[x]$ is given by Tweedie's formula [42, 43]

$$\mathbb{E}_{p(x \mid x(t))}[x] = \frac{x(t) + \sigma(t)^2 \, \nabla_{x(t)} \log p(x(t))}{\mu(t)} \tag{9}$$

$$\approx \frac{x(t) + \sigma(t)^2 \, s_\phi(x(t), t)}{\mu(t)} . \tag{10}$$

As the log-likelihood of a multivariate Gaussian is known analytically and $s_\phi(x(t), t)$ is differentiable, we can compute the likelihood score $\nabla_{x(t)} \log p(y \mid x(t))$ with this approximation in zero-shot, that is, without training any other network than $s_\phi(x(t), t)$.

## 3   Score-based data assimilation

Coming back to our initial inference problem, we want to approximate the trajectory posterior $p(x_{1:L} \mid y)$ of a dynamical system. To do so with score-based generative modeling, we need to estimate the posterior score $\nabla_{x_{1:L}(t)} \log p(x_{1:L}(t) \mid y)$, which we choose to decompose into prior and likelihood terms, as in (7), to enable a wide range of zero-shot observation scenarios.

In typical data assimilation settings, the high-dimensionality of each state $x_i$ (e.g. the state of atmospheres or oceans) combined with potentially long trajectories would require an impractically large score network $s_\phi(x_{1:L}(t), t)$ to estimate the prior score $\nabla_{x_{1:L}(t)} \log p(x_{1:L}(t))$ and a proportional amount of data for training, which could be prohibitive if data is scarce or if the physical model is expensive to simulate. To overcome this challenge, we leverage the Markovian structure of dynamical systems to approximate the prior score with a series of local scores, which are easier to learn, as explained in Section 3.1. In Section 3.2, we build upon diffusion posterior sampling (DPS) [41] to propose a new approximation for the likelihood score $\nabla_{x_{1:L}(t)} \log p(y \mid x_{1:L}(t))$, which we find more appropriate for posterior inference. Finally, in Section 3.3, we describe our sampling procedure inspired from predictor-corrector sampling [28]. Our main contribution, named score-based data assimilation (SDA), is the combination of these three components.

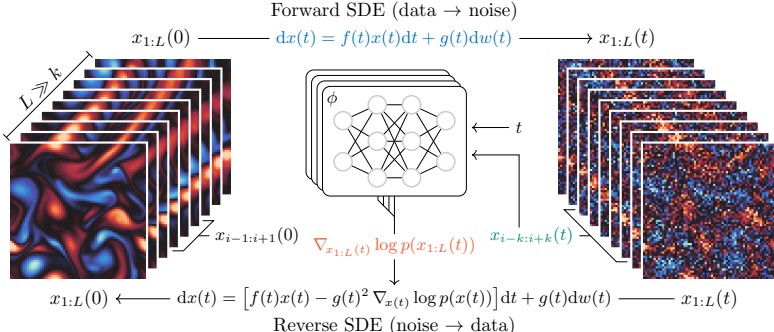

Figure 1: Trajectories $x_{1:L}$ of a dynamical system are transformed to noise via a diffusion process. Reversing this process generates new trajectories, but requires the score of $p(x_{1:L}(t))$. We approximate it by combining the outputs of a score network over sub-segments of $x_{1:L}(t)$.

### 3.1 How is your blanket?

Given a set of random variables $x_{1:L} = \{x_1, x_2, \dots, x_L\}$, it is sometimes possible to find a small Markov blanket $x_{b_i} \subseteq x_{\neq i}$ such that $p(x_i \mid x_{\neq i}) = p(x_i \mid x_{b_i})$ for each element $x_i$ using our knowledge of the set's structure. It follows that each element $\nabla_{x_i} \log p(x_{1:L})$ of the full score $\nabla_{x_{1:L}} \log p(x_{1:L})$ can be determined locally, that is, only using its blanket;

$$\nabla_{x_i} \log p(x_{1:L}) = \nabla_{x_i} \log p(x_i \mid x_{\neq i}) + \nabla_{x_i} \log p(x_{\neq i}) \tag{11}$$
$$= \nabla_{x_i} \log p(x_i \mid x_{b_i}) + \nabla_{x_i} \log p(x_{b_i}) = \nabla_{x_i} \log p(x_i, x_{b_i}). \tag{12}$$

This property generally does not hold for the diffusion-perturbed set $x_{1:L}(t)$ as there is no guarantee that $x_{b_i}(t)$ is a Markov blanket of the element $x_i(t)$. However, there exists a set of indices $\bar{b}_i \supseteq b_i$ such that

$$\nabla_{x_i(t)} \log p(x_{1:L}(t)) \approx \nabla_{x_i(t)} \log p(x_i(t), x_{\bar{b}_i}(t)) \tag{13}$$

is a good approximation for all $t \in [0, 1]$. That is, $x_{\bar{b}_i}(t)$ is a "pseudo" Markov blanket of $x_i(t)$. In the worst case, $\bar{b}_i$ contains all indices except $i$, but we argue that, for some structures, there is a set $\bar{b}_i$ not much larger than $b_i$ that satisfies (13). Our rationale is that, since we impose the initial noise to be negligible, we know that $x_{b_i}(t)$ becomes indistinguishable from $x_{b_i}$ as $t$ approaches 0. Furthermore, as $t$ grows and noise accumulates, the mutual information between elements $x_i(t)$ and $x_j(t)$ decreases to finally reach 0 when $t = 1$. Hence, even if $\bar{b}_i = b_i$, the pseudo-blanket approximation (13) already holds near $t = 0$ and $t = 1$. In between, even though the approximation remains unbiased (see Appendix A), the structure of the set becomes decisive. If it is known and present enough regularities/symmetries, (13) could and should be exploited within the architecture of the score network $s_\phi(x_{1:L}(t), t)$.

In the case of dynamical systems, the set $x_{1:L}$ is by definition a first-order Markov chain and the minimal Markov blanket of an element $x_i$ is $x_{b_i} = \{x_{i-1}, x_{i+1}\}$. For the perturbed element $x_i(t)$, the pseudo-blanket $x_{\bar{b}_i}(t)$ can take the form of a window surrounding $x_i(t)$, that is $\bar{b}_i = \{i - k, \dots, i + k\} \setminus \{i\}$ with $k \geq 1$. The value of $k$ is dependent on the problem, but we argue, supported by our experiments, that it is generally much smaller than the chain's length $L$. Hence, a fully convolutional neural network (FCNN) with a narrow receptive field is well suited to the task, and any long-range capabilities would be wasted resources. Importantly, if the receptive field is $2k + 1$, the network can be trained on segments $x_{i-k:i+k}$ instead of the full chain $x_{1:L}$, thereby drastically reducing training costs. More generally, we can train a local score network (see Algorithm 1)

$$s_\phi(x_{i-k:i+k}(t), t) \approx \nabla_{x_{i-k:i+k}(t)} \log p(x_{i-k:i+k}(t)) \tag{14}$$

such that its $k + 1$-th element approximates the score of the $i$-th state $\nabla_{x_i(t)} \log p(x_{1:L}(t))$. We also have that the $k$ first elements of $s_\phi(x_{1:2k+1}(t), t)$ approximate the score of the $k$ first states $\nabla_{x_{1:k}(t)} \log p(x_{1:L}(t))$ and the $k$ last elements of $s_\phi(x_{L-2k:L}(t), t)$ approximate the score of the $k$ last states $\nabla_{x_{L-k:L}(t)} \log p(x_{1:L}(t))$. Hence, we can apply the local score network on all sub-segments $x_{i-k:i+k}(t)$ of $x_{1:L}(t)$, similar to a convolution kernel, and combine the outputs (see Algorithm 2) to get an approximation of the full score $\nabla_{x_{1:L}(t)} \log p(x_{1:L}(t))$. Note that we can either condition the score network with $i$ or assume the statistical stationarity of the chain, that is $p(x_i) = p(x_{i+1})$.

**Algorithm 1** Training $\epsilon_\phi(x_{i-k:i+k}(t), t)$

1  **for** $i = 1$ to $N$ **do**
2      $x_{1:L} \sim p(x_{1:L})$
3      $i \sim \mathcal{U}(\{k+1, \ldots, L-k\})$
4      $t \sim \mathcal{U}(0,1), \epsilon \sim \mathcal{N}(0, I)$
5      $x_{i-k:i+k}(t) \leftarrow \mu(t)\, x_{i-k:i+k} + \sigma(t)\, \epsilon$
6      $\ell \leftarrow \|\epsilon_\phi(x_{i-k:i+k}(t), t) - \epsilon\|_2^2$
7      $\phi \leftarrow \text{GRADIENTDESCENT}(\phi, \nabla_\phi \ell)$

**Algorithm 2** Composing $s_\phi(x_{i-k:i+k}(t), t)$

1  **function** $s_\phi(x_{1:L}(t), t)$
2      $s_{1:k+1} \leftarrow s_\phi(x_{1:2k+1}(t), t)[:k+1]$
3      **for** $i = k+2$ to $L-k-1$ **do**
4         $s_i \leftarrow s_\phi(x_{i-k:i+k}(t), t)[k+1]$
5      $s_{L-k:L} \leftarrow s_\phi(x_{L-2k:L}(t), t)[k+1:]$
6      **return** $s_{1:L}$

## 3.2 Stable likelihood score

Due to approximation and numerical errors in $\hat{x}(x(t))$, computing the score $\nabla_{x(t)} \log p(y \mid x(t))$ with the likelihood approximation (8) is very unstable, especially in the low signal-to-noise regime, that is when $\sigma(t) \gg \mu(t)$. This incites Chung et al. [41] to replace the covariance $\Sigma_y$ by the identity $I$ and rescale the likelihood score with respect to $\|y - \mathcal{A}(\hat{x}(x(t)))\|$ to stabilize the sampling process. These modifications introduce a significant error in the approximation as they greatly affect the norm of the likelihood score.

We argue that the instability is due to (8) being only exact if the variance of $p(x \mid x(t))$ is null or negligible, which is not the case when $t > 0$. Instead, Adam et al. [40] and Meng et al. [44] approximate the covariance of $p(x \mid x(t))$ with $\Sigma(t)/\mu(t)^2$, which is valid as long as the prior $p(x)$ is Gaussian with a large diagonal covariance $\Sigma_x$. We motivate in Appendix B the more general covariance approximation $\sigma(t)^2/\mu(t)^2 \Gamma$, where the matrix $\Gamma$ depends on the eigendecomposition of $\Sigma_x$. Then, taking inspiration from the extended Kalman filter, we approximate the perturbed likelihood as

$$p(y \mid x(t)) \approx \mathcal{N}\left(y \mid \mathcal{A}(\hat{x}(x(t))), \Sigma_y + \frac{\sigma(t)^2}{\mu(t)^2} A\Gamma A^T\right) \tag{15}$$

where $A = \partial_x \mathcal{A} \mid_{\hat{x}(x(t))}$ is the Jacobian of $\mathcal{A}$. In practice, to simplify the approximation, the term $A\Gamma A^T$ can often be replaced by a constant (diagonal) matrix. We find that computing the likelihood score $\nabla_{x(t)} \log p(y \mid x(t))$ with this new approximation (see Algorithm 3) is stable enough that rescaling it or ignoring $\Sigma_y$ is unnecessary.

## 3.3 Predictor-Corrector sampling

To simulate the reverse SDE, we adopt the exponential integrator (EI) discretization scheme introduced by Zhang et al. [30]

$$x(t - \Delta t) \leftarrow \frac{\mu(t - \Delta t)}{\mu(t)} x(t) + \left(\frac{\mu(t - \Delta t)}{\mu(t)} - \frac{\sigma(t - \Delta t)}{\sigma(t)}\right) \sigma(t)^2\, s_\phi(x(t), t) \tag{16}$$

which coincides with the deterministic DDIM [36] sampling algorithm when the variance preserving SDE [32] is used. However, as we approximate both the prior and likelihood scores, errors accumulate along the simulation and cause it to diverge, leading to low-quality samples. To prevent errors from accumulating, we perform (see Algorithm 4) a few steps of Langevin Monte Carlo (LMC) [45, 46]

$$x(t) \leftarrow x(t) + \delta\, s_\phi(x(t), t) + \sqrt{2\delta}\, \epsilon \tag{17}$$

where $\epsilon \sim \mathcal{N}(0, I)$, between each step of the discretized reverse SDE (16). In the limit of an infinite number of LMC steps with a sufficiently small step size $\delta \in \mathbb{R}_+$, simulated samples are guaranteed to follow the distribution implicitly defined by our approximation of the posterior score at each time $t$, meaning that the errors introduced by the pseudo-blanket (13) and likelihood (15) approximations do not accumulate. In practice, we find that few LMC steps are necessary. Song et al. [28] introduced a similar strategy, named predictor-corrector (PC) sampling, to correct the errors introduced by the discretization of the reverse SDE.

# 4 Results

We demonstrate the effectiveness of score-based data assimilation on two chaotic dynamical systems: the Lorenz 1963 [47] and Kolmogorov flow [48] systems. The former is a simplified mathematical model for atmospheric convection. Its low dimensionality enables posterior inference using classical sequential Monte Carlo methods [49, 50] such as the bootstrap particle filter [51]. This allows us to compare objectively our posterior approximations against the ground-truth posterior. The second system considers the state of a two-dimensional turbulent fluid subject to Kolmogorov forcing [48]. The evolution of the fluid is modeled by the Navier-Stokes equations, the same equations that underlie the models of oceans and atmospheres. This task provides a good understanding of how SDA would perform in typical data assimilation applications, although our analysis is primarily qualitative due to the unavailability of reliable assessment tools for systems of this scale.

For both systems, we employ as diffusion process the variance preserving SDE with a cosine schedule [52], that is $\mu(t) = \cos(\omega t)^2$ with $\omega = \arccos \sqrt{10^{-3}}$ and $\sigma(t) = \sqrt{1 - \mu(t)^2}$. The score networks are trained once and then evaluated under various observation scenarios. Unless specified otherwise, we estimate the posterior score according to Algorithm 3 with $\Gamma = 10^{-2}I$ and simulate the reverse SDE (4) according to Algorithm 4 in 256 evenly spaced discretization steps.

## 4.1 Lorenz 1963

The state $x = (a, b, c) \in \mathbb{R}^3$ of the Lorenz system evolves according to a system of ordinary differential equations

$$\dot{a} = \sigma(b - a)$$
$$\dot{b} = a(\rho - c) - b \tag{18}$$
$$\dot{c} = ab - \beta c$$

where $\sigma = 10$, $\rho = 28$ and $\beta = \frac{8}{3}$ are parameters for which the system exhibits a chaotic behavior. We denote $\tilde{a}$ and $\tilde{c}$ the standardized (zero mean and unit variance) versions of $a$ and $c$, respectively. As our approach assumes a discrete-time stochastic dynamical system, we consider a transition process of the form $x_{i+1} = \mathcal{M}(x_i) + \eta$, where $\mathcal{M} : \mathbb{R}^3 \mapsto \mathbb{R}^3$ is the integration of the differential equations (18) for $\Delta = 0.025$ time units and $\eta \sim \mathcal{N}(0, \Delta I)$ represents Brownian noise.

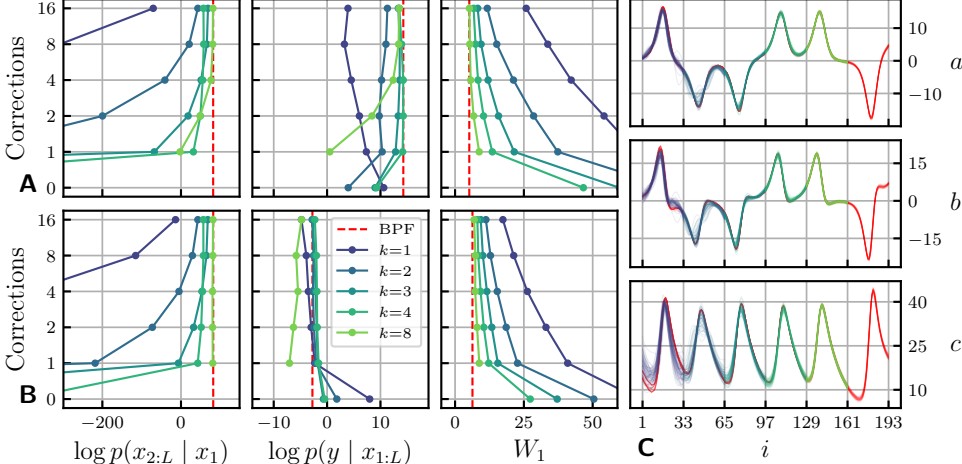

Figure 2: Average posterior summary statistics over 64 observations from the low (**A**) and high (**B**) frequency observation processes. We observe that, as $k$ and the number of corrections $C$ increase, the statistics of the approximate posteriors get closer to the ground-truth, in red, which means they are getting more accurate. However, increasing $k$ and $C$ improves the quality of posteriors with decreasing return, such that all posteriors with $k \geq 3$ and $C \geq 2$ are almost equivalent. This is visible in **C**, where we display trajectories inferred ($C = 2$) for an observation of the low frequency observation process. For readability, we allocate a segment of 32 states to each $k$ instead of overlapping all 192 states. Note that the Wasserstein distance between the ground-truth posterior and itself is not zero as it is estimated with a finite number (1024) of samples.

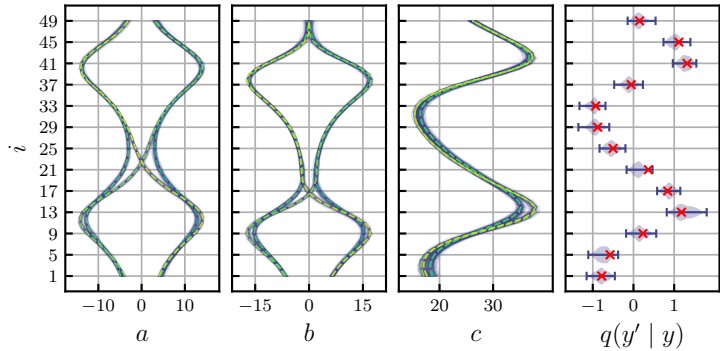

Figure 3: Example of multi-modal posterior inference with SDA. We identify four modes (dashed lines) in the inferred posterior. All modes are consistent with the observation (red crosses), as demonstrated by the posterior predictive distribution $q(y' \mid y) = \mathbb{E}_{q(x_{1:L}|y)}[p(y' \mid x_{1:L})]$.

We generate 1024 independent trajectories of 1024 states, which are split into training ($80\,\%$), validation ($10\,\%$) and evaluation ($10\,\%$) sets. The initial states are drawn from the statistically stationary regime of the system. We consider two score network architectures: fully-connected local score networks for small $k$ ($k \leq 4$) and fully-convolutional score networks for large $k$. Architecture and training details for each $k$ are provided in Appendix D.

We first study the impact of $k$ (see Section 3.1) and the number of LMC corrections (see Section 3.3) on the quality of the inferred posterior. We consider two simple observation processes $\mathcal{N}(y \mid \tilde{a}_{1:L:8}, 0.05^2 I)$ and $\mathcal{N}(y \mid \tilde{a}_{1:L}, 0.25^2 I)$. The former observes the state at low frequency (every eighth step) with low noise, while the latter observes the state at high frequency (every step) with high noise. For both processes, we generate an observation $y$ for a trajectory of the evaluation set (truncated at $L = 65$) and apply the bootstrap particle filter (BPF) to draw 1024 trajectories $x_{1:L}$ from the ground-truth posterior $p(x_{1:L} \mid y)$. We use a large number of particles ($2^{16}$) to ensure convergence. Then, using SDA, we sample 1024 trajectories from the approximate posterior $q(x_{1:L} \mid y)$ defined by each score network. We compare the approximate and ground-truth posteriors with three summary statistics: the expected log-prior $\mathbb{E}_{q(x_{1:L}|y)}[\log p(x_{2:L} \mid x_1)]$, the expected log-likelihood $\mathbb{E}_{q(x_{1:L}|y)}[\log p(y \mid x_{1:L})]$ and the Wasserstein distance $W_1(p, q)$ in trajectory space. We repeat the procedure for 64 observations and different number of corrections ($\tau = 0.25$, see Algorithm 4) and present the results in Figure 2. To paraphrase, SDA is able to reproduce the ground-truth posterior accurately. Interestingly, accuracy can be traded off for computational efficiency: fewer corrections leads to faster inference at the potential expense of physical consistency.

Another advantage of SDA over variational data assimilation approaches is that it targets the whole posterior distribution instead of point estimates, which allows to identify when several scenarios are plausible. As a demonstration, we generate an observation from the observation process $p(y \mid x_{1:L}) = \mathcal{N}(y \mid \tilde{c}_{1:L:4}, 0.1^2 I)$ and infer plausible trajectories with SDA ($k = 4$, $C = 2$). Several modes are identified in the posterior, which we illustrate in Figure 3.

## 4.2 Kolmogorov flow

Incompressible fluid dynamics are governed by the Navier-Stokes equations

$$\dot{\boldsymbol{u}} = -\boldsymbol{u}\nabla\boldsymbol{u} + \frac{1}{Re}\nabla^2\boldsymbol{u} - \frac{1}{\rho}\nabla p + \boldsymbol{f}$$
$$0 = \nabla \cdot \boldsymbol{u}$$

(19)

where $\boldsymbol{u}$ is the velocity field, $Re$ is the Reynolds number, $\rho$ is the fluid density, $p$ is the pressure field and $\boldsymbol{f}$ is the external forcing. Following Kochkov et al. [53], we choose a two-dimensional domain $[0, 2\pi]^2$ with periodic boundary conditions, a large Reynolds number $Re = 10^3$, a constant density $\rho = 1$ and an external forcing $\boldsymbol{f}$ corresponding to Kolmogorov forcing with linear damping [48, 54]. We use the `jax-cfd` library [53] to solve the Navier-Stokes equations (19) on a $256 \times 256$ domain grid. The states $x_i$ are snapshots of the velocity field $\boldsymbol{u}$, coarsened to a $64 \times 64$ resolution, and the integration time between two such snapshots is $\Delta = 0.2$ time units. This corresponds to 82

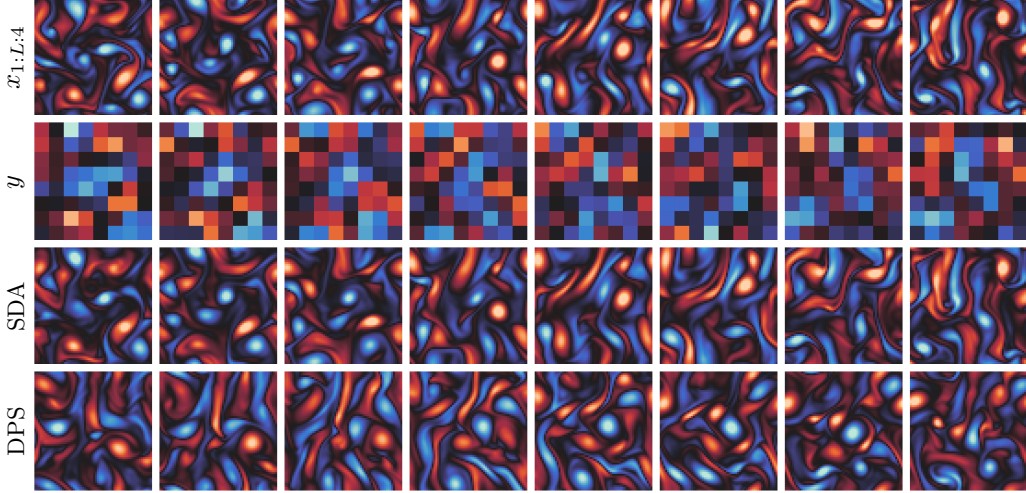

Figure 4: Example of sampled trajectory from coarse, intermittent and noisy observations. States are visualized by their vorticity field $\omega = \nabla \times \boldsymbol{u}$, that is the curl of the velocity field. Positive values (red) indicate clockwise rotation and negative values (blue) indicate counter-clockwise rotation. SDA closely recovers the original trajectory, despite the limited amount of available data. Replacing SDA's likelihood score approximation with the one of DPS [41] yields trajectories inconsistent with the observation.

integration steps of the forward Euler method, which would be expensive to differentiate through repeatedly, as required by gradient-based data assimilation approaches.

We generate 1024 independent trajectories of 64 states, which are split into training (80 %), validation (10 %) and evaluation (10 %) sets. The initial states are drawn from the statistically stationary regime of the system. We consider a local score network with $k = 2$. As states take the form of $64 \times 64$ images with two velocity channels, we use a U-Net [55] inspired network architecture. Architecture and training details are provided in Appendix D.

We first apply SDA to a classic data assimilation problem. We take a trajectory of length $L = 32$ from the evaluation set and observe the velocity field every four steps, coarsened to a resolution $8 \times 8$ and perturbed by a moderate Gaussian noise ($\Sigma_y = 0.1^2 I$). Given the observation, we sample a trajectory with SDA ($C = 1$, $\tau = 0.5$) and find that it closely recovers the original trajectory, as illustrated in Figure 4. A similar experiment where we modify the amount of spatial information is presented in Figure 8. When data is insufficient to identify the original trajectory, SDA extrapolates a physically plausible scenario while remaining consistent with the observation, which can also be observed in Figure 6 and 7.

$$\xleftarrow{\hspace{3cm}} \text{SDA} \xrightarrow{\hspace{3cm}} y = \mathcal{A}(x_L) + \eta$$

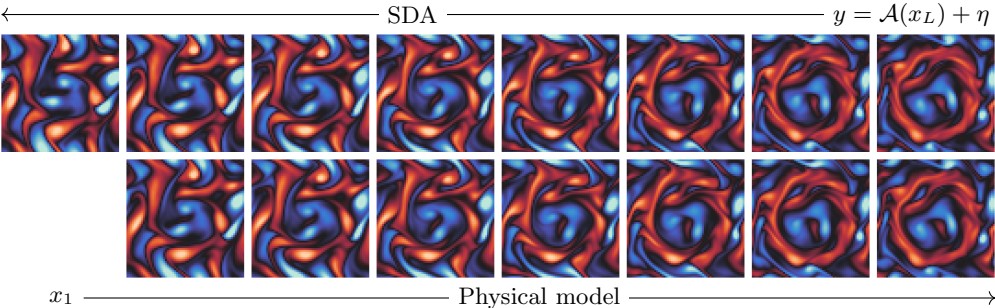

$$x_1 \xrightarrow{\hspace{3cm}} \text{Physical model} \xrightarrow{\hspace{3cm}}$$

Figure 5: A trajectory consistent with an unlikely observation of the final state $x_L$ is generated with SDA. To verify whether the trajectory is realistic and not hallucinated, we plug its initial state $x_1$ into the physical model and obtain an almost identical trajectory. This indicates that SDA is not simply interpolating between observations, but rather propagates information in a manner consistent with the physical model, even in unlikely scenarios.

Finally, we investigate whether SDA generalizes to unlikely scenarios. We design an observation process that probes the vorticity of the final state $x_L$ in a circle-shaped sub-domain. Then, we sample a trajectory ($C = 1$, $\tau = 0.5$) consistent with a uniform positive vorticity observation in this sub-domain, which is unlikely, but not impossible. The result is discussed in Figure 5.

## 5 Conclusion

**Impact**   In addition to its contributions to the field of data assimilation, this work presents new technical contributions to the field of score-based generative modeling.

First, we provide new insights on how to exploit conditional independences (Markov blankets) in sets of random variables to build and train score-based generative models. Based on these findings, we are able to generate/infer simultaneously all the states of arbitrarily long Markov chains $x_{1:L}$, while only training score models on short segments $x_{i-k:i+k}$, thereby reducing the training costs and the amounts of training data required. The decomposition of the global score into local scores additionally allows for better parallelization at inference, which could be significant depending on available hardware. Importantly, the pseudo-blanket approximation (13) is not limited to Markov chains, but could be applied to any set of variables $x_{1:L}$, as long as some structure is known.

Second, we motivate and introduce a novel approximation (15) for the perturbed likelihood $p(y|x(t))$, when the likelihood $p(y|x)$ is assumed (linear or non-linear) Gaussian. We find that computing the likelihood score $\nabla_{x(t)} \log p(y|x(t))$ with this new approximation leads to accurate posterior inference, without the need for stability tricks [41]. This contribution can be trivially adapted to many tasks such as inpainting, deblurring, super-resolution or inverse problems in scientific fields [39–41].

**Limitations**   From a computational perspective, even though SDA does not require simulating or differentiating through the physical model, inference remains limited by the speed of the simulation of the reverse SDE. Accelerating sampling in score-based generative models is an active area of research [30, 36, 37, 56] with promising results which would be worth exploring in the context of our method.

Regarding the quality of our results, we empirically demonstrate that SDA provides accurate approximations of the whole posterior, especially as $k$ and the number of LMC corrections $C$ increase. However, our approximations (13) and (15) introduce a certain degree of error, which precise impact on the resulting posterior remains to be theoretically quantified. Furthermore, although the Kolmogorov system is high-dimensional (tens of thousands of dimensions) with respect to what is approachable with classical posterior inference methods, it remains small in comparison to the millions of dimensions of some operational DA systems. Whether SDA would scale well to such applications is an open question and will present serious engineering challenges.

Another limitation of our work is the assumption that the dynamics of the system are shared by all trajectories. In particular, if a parametric physical model is used, all trajectories are assumed to share the same parameters. For this reason, SDA is not applicable to settings where fitting the model parameters is also required, or at least not without further developments. Some approaches [57–60] tackle this task, but they remain limited to low-dimensional settings. Additionally, if a physical model is used to generate synthetic training data, instead of relying on real data, one can only expect SDA to be as accurate as the model itself. This is a limitation shared by any model-based approach and robust assimilation under model misspecification or distribution shift is left as an avenue for future research.

Finally, posterior inference over entire state trajectories is not always necessary. In forecasting tasks, inferring the current state of the dynamical system is sufficient and likely much less expensive. In this setting, data assimilation reduces to a state estimation problem for which classical methods such as the Kalman filter [61] or its nonlinear extensions [62, 63] provide strong baselines. Many deep learning approaches have also been proposed to bypass the physical model entirely and learn instead a generative model of plausible forecasts from past observations only [64–67].

**Related work**   A number of previous studies have investigated the use of deep learning to improve the quality and efficiency of data assimilation. Mack et al. [12] use convolutional auto-encoders to project the variational data assimilation problem into a lower-dimensional space, which simplifies the optimization problem greatly. Frerix et al. [14] use a deep neural network to predict the initial state of a trajectory given the observations. This prediction is then used as a starting point for traditional

(4D-Var) variational data assimilation methods, which proves to be more effective than starting at random. This strategy is also possible with SDA (using a trajectory sampled with SDA as a starting point) and could help cover multiple modes of the posterior distribution. Finally, Brajard et al. [17] address the problem of simultaneously learning the transition dynamics and estimating the trajectory, when only the observation process is known.

Beyond data assimilation, SDA closely relates to the broader category of sequence models, which have been studied extensively for various types of data, including text, audio, and video. The latest advances demonstrate that score-based generative models achieve remarkable results on the most demanding tasks. Kong et al. [26] and Goel et al. [27] use score-based models to generate long audio sequences non-autoregressively. Ho et al. [23] train a score-based generative model for a fixed number of video frames and use it autoregressively to generate videos of arbitrary lengths. Conversely, our approach is non-autoregressive which allows to generate and condition all elements (frames) simultaneously. Interestingly, as part of their method, Ho et al. [23] introduce "reconstruction guidance" for conditional sampling, which can be seen as a special case of our likelihood approximation (15) where the observation $y$ is a subset of $x$. Lastly, Ho et al. [25] generate low-frame rate, low-resolution videos which are then up-sampled temporally and spatially with a cascade [24] of super-resolution diffusion models. The application of this approach to data assimilation could be worth exploring, although the introduction of arbitrary observation processes seems challenging.

## Acknowledgments and Disclosure of Funding

François Rozet is a research fellow of the F.R.S.-FNRS (Belgium) and acknowledges its financial support.

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

# A The pseudo-blanket approximation is unbiased

For any continuous random variables $a$, $b$ and $c$,

$$
\begin{aligned}
\nabla_a \log p(a, b) &= \frac{1}{p(a, b)} \nabla_a \, p(a, b) \\
&= \frac{1}{p(a, b)} \nabla_a \int p(a, b, c) \, \mathrm{d}c \\
&= \int \frac{p(c \mid a, b)}{p(a, b, c)} \nabla_a \, p(a, b, c) \, \mathrm{d}c \\
&= \mathbb{E}_{p(c \mid a, b)} \left[ \nabla_a \log p(a, b, c) \right] .
\end{aligned}
$$

Replacing $a$, $b$ and $c$ by $x_i(t)$, $x_{\bar{b}_i}(t)$ and $x_{\not\in}(t) = \{x_j(t) : j \neq i \wedge j \notin \bar{b}_i\}$, respectively, we obtain

$$
\nabla_{x_i(t)} \log p(x_i(t), x_{\bar{b}_i}(t)) = \mathbb{E}_{p(x_{\not\in}(t) \mid x_i(t), x_{\bar{b}_i}(t))} \left[ \nabla_{x_i(t)} \log p(x_{1:L}(t)) \right] ,
$$

meaning that $\nabla_{x_i(t)} \log p(x_i(t), x_{\bar{b}_i}(t))$ is the expected value of $\nabla_{x_i(t)} \log p(x_{1:L}(t))$ over $x_{\not\in}(t)$. In other words, regardless of the elements or the size of $\bar{b}_i$, $\nabla_{x_i(t)} \log p(x_i(t), x_{\bar{b}_i}(t))$ is an unbiased (exact in expectation) estimate of $\nabla_{x_i(t)} \log p(x_{1:L}(t))$.

# B On the covariance of $p(x \mid x(t))$

Assuming a Gaussian prior $p(x)$ with covariance $\Sigma_x$, the covariance $\hat{\Sigma}$ of $p(x \mid x(t))$ takes the form

$$
\begin{aligned}
\hat{\Sigma} &= \Sigma_x - \Sigma_x \left( \Sigma_x + \frac{\sigma(t)^2}{\mu(t)^2} I \right)^{-1} \Sigma_x \\
&= \frac{\sigma(t)^2}{\mu(t)^2} Q \Lambda \left( \Lambda + \frac{\sigma(t)^2}{\mu(t)^2} I \right)^{-1} Q^{-1}
\end{aligned}
$$

where $Q \Lambda Q^{-1}$ is the eigendecomposition of $\Sigma_x$. We observe that for most of the perturbation time $t$, the central diagonal term is close to $\Lambda (\Lambda + I)^{-1}$. We therefore propose the covariance approximation

$$
\hat{\Sigma} = \frac{\sigma(t)^2}{\mu(t)^2} \Gamma
$$

where $\Gamma = Q \Lambda \left( \Lambda + I \right)^{-1} Q^{-1}$ is a positive semi-definite matrix.

# C Algorithms

---

**Algorithm 3** Estimating the posterior score $\nabla_{x(t)} \log p(x(t) \mid y)$

---

1 **function** $s_\phi(x(t), t \mid y)$
2      $s_x \leftarrow s_\phi(x(t), t)$
3      $\hat{x} \leftarrow \frac{x(t) + \sigma(t)^2 \, s_x}{\mu(t)}$
4      $s_y \leftarrow \nabla_{x(t)} \log \mathcal{N}\left(y \mid \mathcal{A}(\hat{x}), \Sigma_y + \frac{\sigma(t)^2}{\mu(t)^2}\Gamma\right)$
5      **return** $s_x + s_y$

---

---

**Algorithm 4** Predictor-Corrector sampling from $s_\phi(x(t), t \mid y)$

---

1 **function** SAMPLE($\{t_i\}_{i=0}^N, C, \tau$)
2      $x(1) \sim \mathcal{N}(0, \Sigma(1))$
3      **for** $i = N$ to $1$ **do**
4          $x(t_{i-1}) \leftarrow \frac{\mu(t_{i-1})}{\mu(t_i)} x(t_i) + \left(\frac{\mu(t_{i-1})}{\mu(t_i)} - \frac{\sigma(t_{i-1})}{\sigma(t_i)}\right) \sigma(t_i)^2 \, s_\phi(x(t_i), t_i \mid y)$
5          **for** $j = 1$ to $C$ **do**
6              $\epsilon \sim \mathcal{N}(0, I)$
7              $s \leftarrow s_\phi(x(t_{i-1}), t_{i-1} \mid y)$
8              $\delta \leftarrow \tau \frac{\dim(s)}{\|s\|_2^2}$
9              $x(t_{i-1}) \leftarrow x(t_{i-1}) + \delta \, s + \sqrt{2\delta} \, \epsilon$
10     **return** $x(0)$

---

# D  Experiment details

**Resources**   Experiments were conducted with the help of a cluster of GPUs. In particular, score networks were trained and evaluated concurrently, each on a single GPU with at least 11 GB of memory.

**Lorenz 1963**   We consider two score network architectures: fully-connected local score networks for small $k$ ($k \leq 4$) and fully-convolutional score networks for large $k$ ($k > 4$). Residual blocks [68], SiLU [69] activation functions and layer normalization [70] are used for both architecture. The number of blocks in the fully-convolutional architecture controls the value of $k$. We train all score networks for 1024 epochs with the AdamW [71] optimizer and a linearly decreasing learning rate. Other hyperparameters are provided in Table 1.

Table 1: Score network hyperparameters for the Lorenz experiment.

| Hyperparameter | $k \leq 4$ | $k > 4$ |
|---|---|---|
| Architecture | fully-connected | fully-convolutional |
| Residual blocks | 5 | $k - 2$ |
| Features/channels | 256 | 64 |
| Kernel size | – | 3 |
| Activation | SiLU | SiLU |
| Normalization | LayerNorm | LayerNorm |
| Optimizer | AdamW | AdamW |
| Weight decay | $10^{-3}$ | $10^{-3}$ |
| Learning rate | $10^{-3}$ | $10^{-3}$ |
| Scheduler | linear | linear |
| Epochs | 1024 | 1024 |
| Batches per epoch | 256 | 256 |
| Batch size | 256 | 64 |

**Kolmogorov flow**   The local score network is a U-Net [55] with residual blocks [68], SiLU [69] activation functions and layer normalization [70]. This architecture is motivated by the locality of the Navier-Stokes equations, which impose that the evolution of a drop of fluid is determined by its immediate environment. This can be seen as an application of the pseudo-blanket approximation (13) to a grid-structured set of random variables. We train the score network for 1024 epochs with the AdamW [71] optimizer and a linearly decreasing learning rate. Other hyperparameters are provided in Table 2.

Table 2: Score network hyperparameters for the Kolmogorov experiment.

| | |
|---|---|
| Architecture | U-Net |
| Residual blocks per level | $(3, 3, 3)$ |
| Channels per level | $(96, 192, 384)$ |
| Kernel size | 3 |
| Padding | circular |
| Activation | SiLU |
| Normalization | LayerNorm |
| Optimizer | AdamW |
| Weight decay | $10^{-3}$ |
| Learning rate | $2 \times 10^{-4}$ |
| Scheduler | linear |
| Epochs | 1024 |
| Batches per epoch | 128 |
| Batch size | 32 |

# E   Assimilation examples

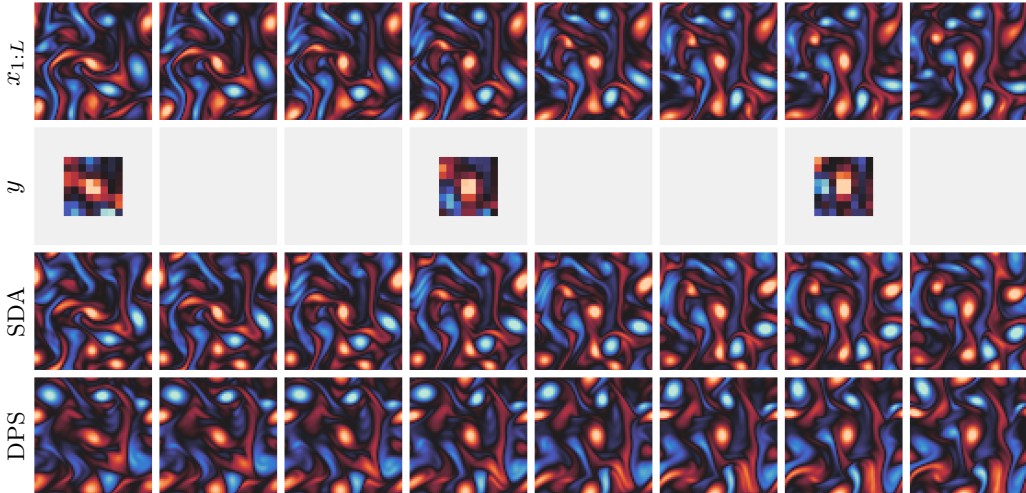

Figure 6: Example of sampled trajectory when the observation is insufficient to identify the original trajectory. The observation is the center of the velocity field every three steps, coarsened to a resolution $8 \times 8$ and perturbed by a small Gaussian noise ($\Sigma_y = 0.01^2 I$). SDA ($C = 1$, $\tau = 0.5$) identifies the original state where data is sufficient, while generating physically plausible states elsewhere. Replacing SDA's likelihood score approximation with the one of DPS [41] yields a trajectory less consistent with the observation.

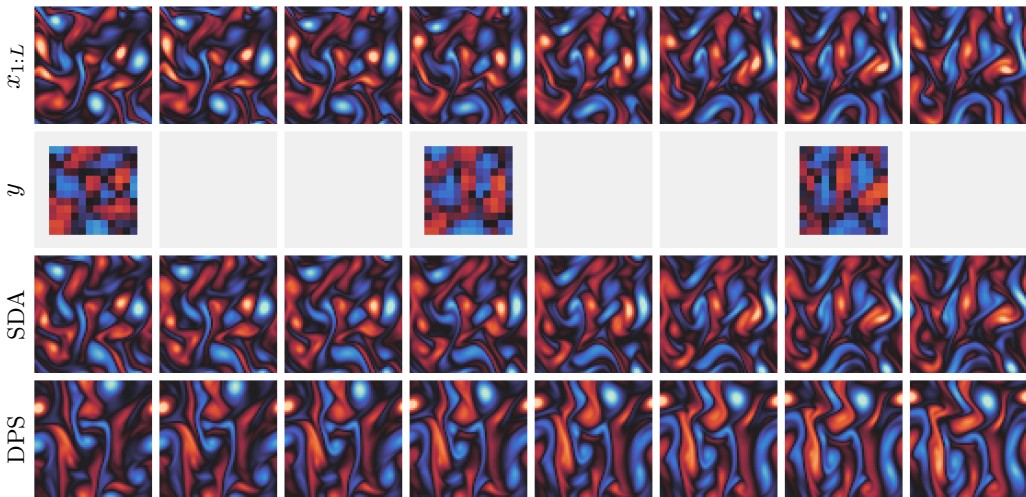

Figure 7: Example of sampled trajectory when the observation process is non-linear. The observation corresponds to a saturating transformation $x \mapsto \frac{x}{1+|x|}$ of the vorticity field $\omega$. SDA (512 discretization steps, $C = 1$, $\tau = 0.5$) identifies the original state where data is sufficient, while generating physically plausible states elsewhere. Replacing SDA's likelihood score approximation with the one of DPS [41] yields a trajectory inconsistent with the observation.

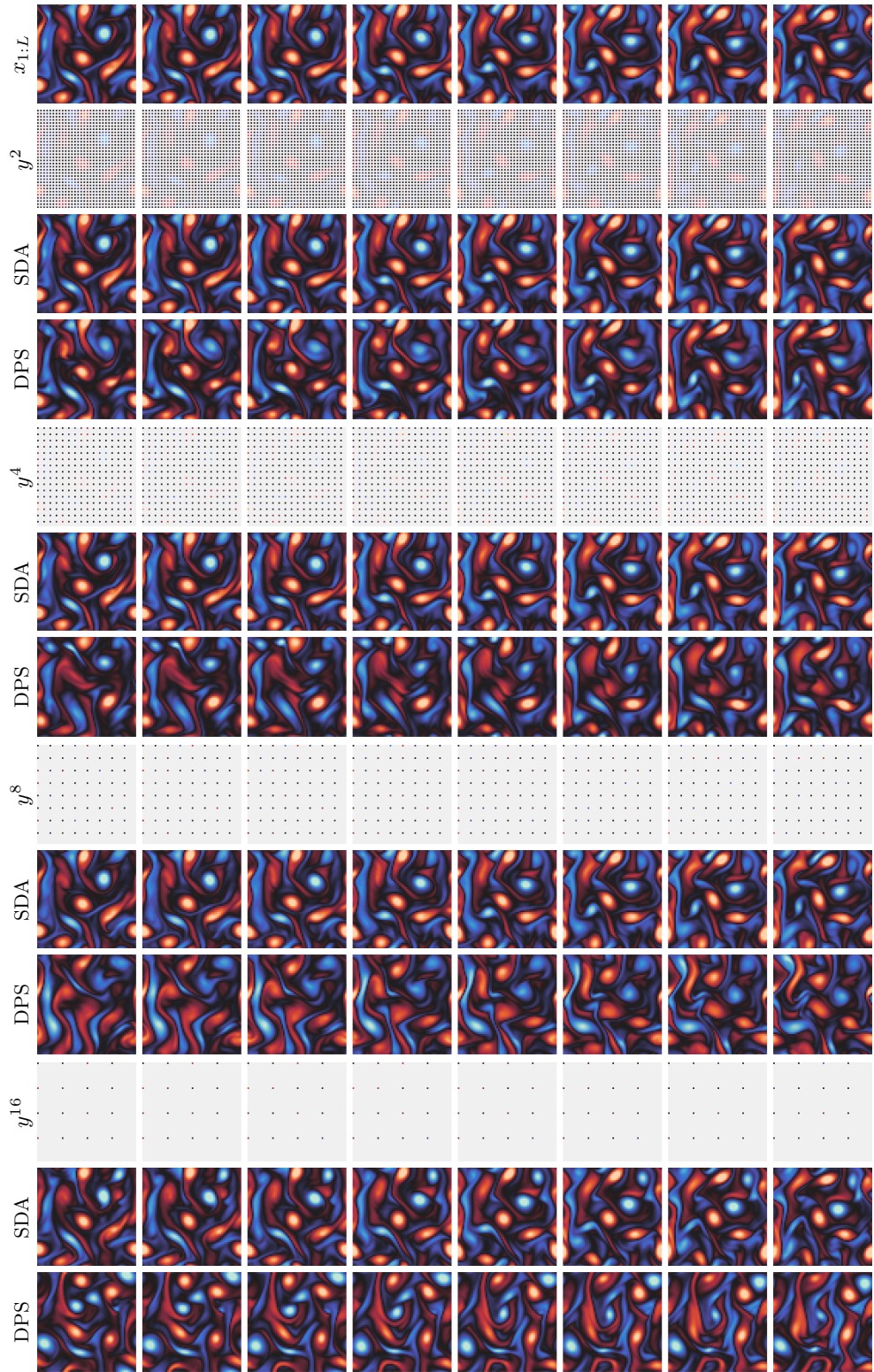

Figure 8: Example of sampled trajectories when the observation is spatially sparse. The observation $y^n$ corresponds to a spatial subsampling of factor $n$ of the velocity field. SDA ($C = 1$, $\tau = 0.5$) closely recovers the original trajectory for all factors $n$, despite the limited amount of available data. Replacing SDA's likelihood score approximation with the one of DPS [41] leads to trajectories that are progressively less consistent with the observation as $n$ increases.

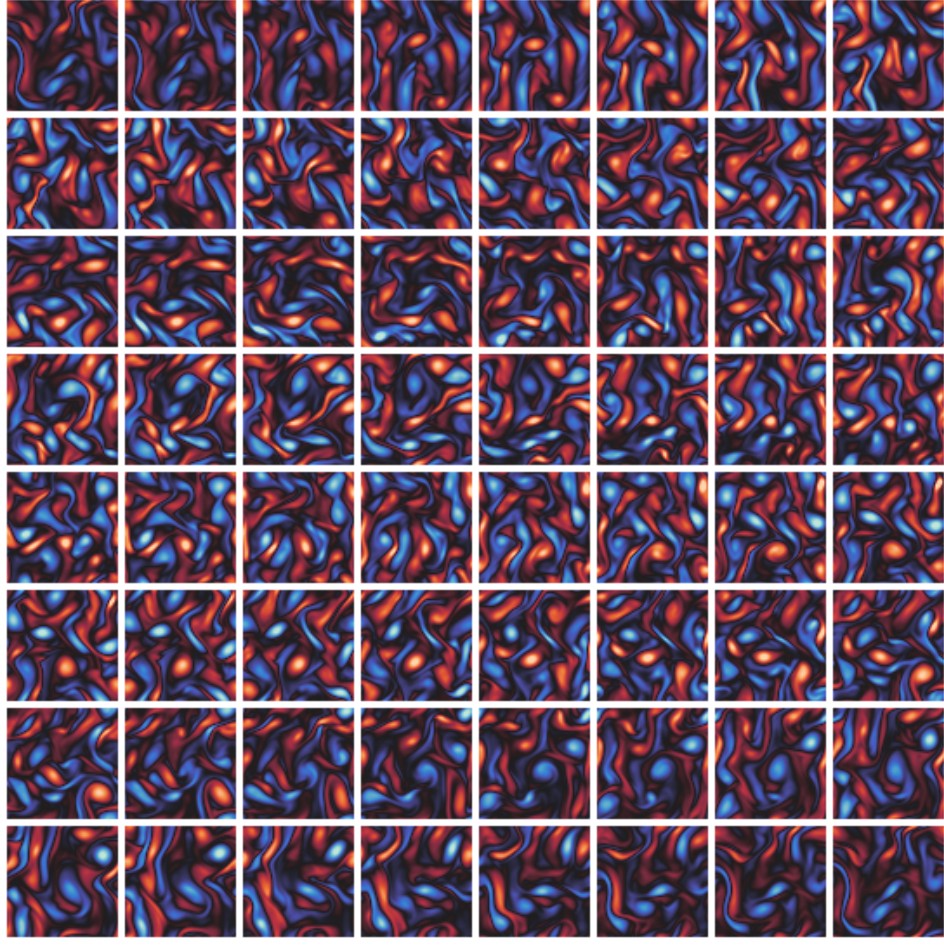

Figure 9: Example of long ($L = 127$) sampled trajectory. Odd states are displayed from left to right and from top to bottom. The observation process probes the difference between the initial and final states and the observation is set to zero, which enforces a looping trajectory ($x_1 \approx x_L$). Note that this scenario is not realistic and will therefore lead to physically implausible trajectories.

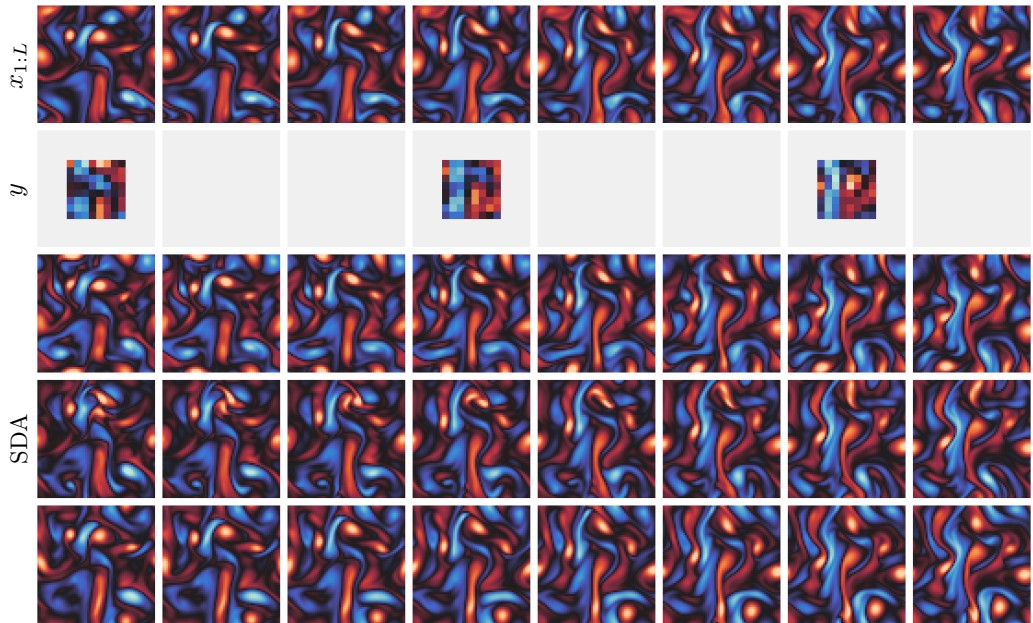

Figure 10: Example of sampled trajectories when the observation is insufficient to identify the original trajectory. The observation is the center of the velocity field every three steps, coarsened to a resolution $8 \times 8$ and perturbed by a small Gaussian noise ($\Sigma_y = 0.01^2 I$). SDA ($C = 1$, $\tau = 0.5$) identifies the original state where data is sufficient, while generating physically plausible states elsewhere.

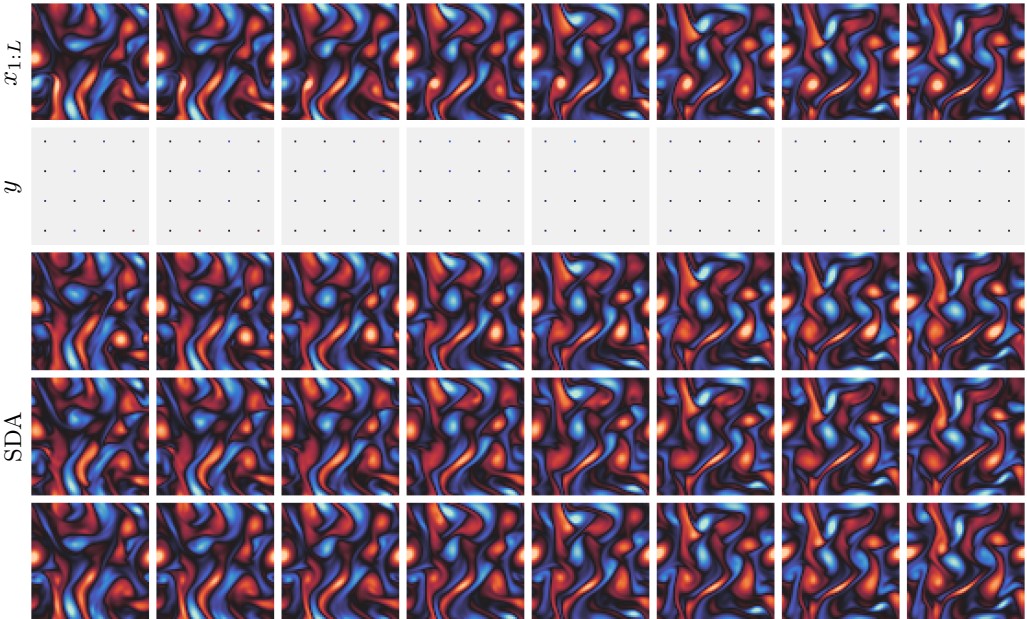

Figure 11: Example of sampled trajectories when the observation is spatially sparse. The observation $y$ corresponds to a spatial subsampling of factor 16 of the velocity field with small Gaussian noise ($\Sigma_y = 0.01^2 I$). SDA ($C = 1$, $\tau = 0.5$) generates trajectories similar to the original, with physically plausible variations.

