# OpenReview forum: "Score-based Data Assimilation"
_NeurIPS.cc/2023/Conference — NeurIPS 2023 poster_

### Official Review · Reviewer_RS1N · 2023-06-27

**Soundness:** 3 good
**Presentation:** 4 excellent
**Contribution:** 4 excellent
**Rating:** 8
**Confidence:** 3

**Summary:**

This paper proposes an approach to solve data-assimilation tasks using techniques from score-based generative modeling. Given observations and trajectories from the data distribution, a posterior distribution over an entire state-trajectory is to be computed. The score function of the posterior is estimated using a learned prior-score network and a Gaussian approximation to the observation model. To make the score-based approach scale to long state trajectories, the authors propose to estimate the prior score over the entire trajectory as a combination of scores computed on local time windows. The theoretical groundwork is backed by a set of experiments in simulated environments.

**Strengths:**

- The presentation of the method is clear and understandable
- This approach to data assimilation is unlike any I have seen before. It combines two established techniques in a natural way without bending one or the other out of shape. The method yielded by said combination appears natural and it seems valuable to test it on real-world applications.
- The authors provide clean code in their supplementary material
- The limitation section answered open questions I had while reading the paper. It lists approximations made and work that is left to do in the future.
- My feeling is that this is the ground work to a fairly interesting and useful approach to inference in dynamical systems

**Weaknesses:**

- From my understanding, the method has yet to be tested on (settings on the scale of) actual real-world problems. Concretely, the experiments shown -- yet convincing -- are rather small-scale, considering that data assimilation regularly has to tackle problems with several millions of state variables. This is not necessarily to be expected when presenting a novel technique, but the Kolmogorov flow experiment left me wondering if there was something standing in the way of choosing a grid that is larger than 64 x 64.
(Note: This is not asking for an additional experiment, but rather my opinion that a statement regarding scalability of the approach would be a good addition to the manuscript.)
- I suspect that -- as is -- the amount of training data that would have to be generated for very large-scale tasks could be computationally quite infeasible

**Questions:**

- Is the approach scalable to massive-scale real-world problems, such as weather forecasting?
  - Is it feasible to generate as many training trajectories as would be needed for such complex dynamics?
- Was the performance of the method studied under dynamics that develop turbulences or even shocks (such as, e.g., Burgers' equation)?

**Limitations:**

- Limitations were stated in the main text.

---

> ### Author Rebuttal · Authors · 2023-08-08
>
> Thank you for your enthusiastic review and for acknowledging the code provided in the supplementary material. We hope that the following answers will satisfy you.
>
> **Weaknesses & Limitations**
>
> > From my understanding, the method has yet to be tested on (settings on the scale of) actual real-world problems. [...] a statement regarding scalability of the approach would be a good addition to the manuscript.
>
> This is a good point. We propose to add the following paragraph in Section 5.
>
> * Concerning our experiments, although the Kolmogorov system is already high-dimensional (tens of thousands of dimensions) with respect to what is approachable with classical posterior inference methods, it remains small in comparison to the millions of dimensions of some operational DA systems. Whether SDA would scale well to such applications is an open question and will obviously present serious engineering challenges.
>
> Nevertheless, we would like to mention that operational DA is usually performed on clusters with hundreds to thousands of CPUs/GPUs, while training and inference in our experiments were performed with a single GPU. We are actively working on/looking for collaborations with weather and oceanic research centers to apply SDA on operational data.
>
> > I suspect that -- as is -- the amount of training data that would have to be generated for very large-scale tasks could be computationally quite infeasible.
>
> In most dynamical systems, many symmetries and inductive biases (e.g. equivariances) can be exploited to reduce the amount of data necessary for training. This is even more true with SDA, as we train on short segments instead of complete trajectories. In fact, for the Kolmogorov system the full training set consists of 820 trajectories of 64 states, which is ridiculously small in comparison to the amounts of data typically used to train image/video generation models.
>
> **Questions**
>
> > Was the performance of the method studied under dynamics that develop turbulences or even shocks (such as, e.g., Burgers' equation)?
>
> The Kolmogorov system considers the state of a two-dimensional turbulent fluid. We did not apply SDA on systems presenting shocks, but this would be an interesting application.

---

> > ### Comment · Reviewer_RS1N · 2023-08-14
> >
> > Many thanks for your response and for acknowledging the open question of scalability in your added paragraph. Based on my current assessment I continue to argue in favor of acceptance and will maintain my initial score.

---

### Official Review · Reviewer_RNbZ · 2023-07-06

**Soundness:** 3 good
**Presentation:** 3 good
**Contribution:** 3 good
**Rating:** 7
**Confidence:** 4

**Summary:**

This paper presents a technique to tackle data assimilation problems (inverse problems involving partially observed time series with a dynamical model acting as a prior over the reconstructed states) using a score based model to estimate the prior distribution $p(x)$ from data. The method used, at its core, leverages a now well-known capability of unconditional score based models to address inverse problems by incorporating the score of the likelihood into the generating process. This makes the approach appealing in zero-shot problems, where a new inverse problem with the same dynamical prior can be solved without any retraining. The paper is the first (to my knowledge) to adapt these techniques for data assimilation. It further proposes a few technical contributions on score based modeling : i) the capacity to exploit the Markov structure of dynamical systems data to train the score based model by small time chunks rather than a global one for the whole trajectories ii) A new approximation technique to estimate the score of the likelihood term $p(y|x(t))$, which is no trivial task iii) and a predictor corrector sample technique to improve data generation by incorporating a few Langevin Monte Carlo sampling iterations in between the steps of the SDE solver. The method is then tested on various assimilation tasks on two datasets coming from the Lorenz-63 system (in a quantitative way since the posterior can be arbitrarily well approximated by a Bootstrap Particle Filter) and Kolmogorov fluid flow (in a qualitative way).


**Strengths:**

- The paper gives a clear exposition of score based methods and their use to solve inverse problems. It also reviews/improves a number of techniques used for efficient training of the model and incorporation of the inverse problem likelihood into the score function.
- This seems to be the first application of score based models for posterior sampling in data assimilation, and it shows nice potential for those applications (emulating the physical model without the need to differentiate through the whole dynamical model and a solver, and reusing it for any assimilation problem using the same dynamics). The applications tackled, although relatively simple (in their dynamics), lead to think this could be applied to more ambitious situations in the future.
- The paper presents three additional technical contributions useful for the generative modeling community as a whole, improving sample generation/training local score based models adapted to Markov chains/improving the approximation of the score of the likelihood. These contributions and the associated claims are globally well supported by the experiments.


**Weaknesses:**

- Section 3.1 should be a bit expanded, I believe. In particular, using the blanket to compute equations (11) and (12) could be a bit more detailed, and the link between the result of appendix A and the unbiasedness of the estimate (13) could be a bit explicited. It would be nice also to have an idea of how the value of k affects training time. However, the idea is sound and the approximation looks okay in the results on the Lorenz system.  So overall, the blanket is good.
- More generally, there is a lot of emphasis in the paper on making the model easier to train/less expensive, less memory consuming or less prone to technical/computational difficulties (e.g. differentiating through the physical model and its associated solver), but none of this is actually quantified. For example, it would be quite interesting to test the effect of the value of $k$ on the training time compared to a naive global model, to quantify the effect of the number of correction steps $C$ on running time instead of just the quality of the solutions.
- In addition, the paper would benefit from the comparison to more standard data assimilation techniques. This is done on the Lorenz data since a BPF is used to sample from the true posterior, but it would be nice to have an idea of the quality of the SDA output with respect to say, a classical variational assimilation with differentiation through the model (both in performance and running time). The training time should be reported, as well as the inference time, to be compared with the computation cost of optimizing a variational assimilation cost function.

**Questions:**

- Can the authors highlight the difference/added value between their predictor-corrector sampler and the one in [24] ? The rationale seems very similar even though the equations differ. Please explain.
- In figure 4, would it be possible to quantify the closeness to the observations quantitatively, since this is the qualitative way of assessing the result on the Kolmogorov flow experiment? And compare it to what a competitor technique would output.
- Similarly, it would be interesting to show several posterior samples on one of these experiments to have an idea of whether the model exhibits a multimodal posterior in this situation (I would expect this e.g. in the super-resolution and extrapolation applications). An analog of figure 3 for the second case study, basically.
- Maybe a more detailed explanation of Tweedie’s formula and the computation of equations (9) and (10) would be nice to make the paper more self-contained.

**Limitations:**

- To me the main limitation is the lack of information about the practical impact of the contributions on the Markov blanket/correction step on the running time/memory in addition to simply their approximation capability. See above for more detailed remarks on this.
- As mentioned by the authors, the dynamical model is assumed to be perfectly known or at the very least fixed. However, this is not true for the observation model, which makes the proposed method versatile as long as one is interested in various inverse problems with the same dynamical model. An interesting next step would be to see how to enable fitting/fine tuning model parameters, but since here the model is never used but rather emulated, this seems like a difficult endeavor. By the way, going through an emulator of the dynamics rather than having to run/differentiate the model could also be an advantage for complex models that cannot be easily made differentiable (as is often the case for operational models), provided they can be emulated successfully by a score based model.
- On a related note, another score based model training is required in case of change in dynamical model (but this is the same for variational data assimilation techniques)

---

> ### Author Rebuttal · Authors · 2023-08-08
>
> Thank you for your in-depth review and feedback. Most of your concerns are sensible and will be addressed in the manuscript.
>
> **Weaknesses & Limitations**
>
> > In particular, using the blanket to compute equations (11) and (12) could be a bit more detailed, and the link between the result of appendix A and the unbiasedness of the estimate (13) could be a bit explicited. [...] Maybe a more detailed explanation of Tweedie’s formula and the computation of equations (9) and (10) would be nice to make the paper more self-contained.
>
> We will motivate and explain these equations more explicitly in the manuscript.
>
> > There is a lot of emphasis in the paper on making the model easier to train/less expensive, less memory consuming or less prone to technical/computational difficulties, but none of this is actually quantified.
>
> Indeed, the primary goal of Section 3.1 is to make the training of the prior score easier/less expensive. If we were to train a (global) score network on trajectories $x_{1:L}$, we would perform amounts of computation proportional to $L$, which could be huge depending on the dimensionality of the states $x_i$. Conversely, training a local score network only requires segments $x_{i-k:i+k}$ instead of complete trajectories $x_{1:L}$. An immediate consequence is that the cost of training is drastically reduced; roughly by a factor $\frac{L}{2k + 1}$. A more subtle consequence is that the decomposition of the global score into local scores imposes a form of translational equivariance, which reduces the amounts of training data required.
>
> We propose to replace (lines 143-144)
>
> "Still, in some settings where each element $x_i$ is high-dimensional, building and training an FCNN for the full chain $x_{1:L}$ could be technically difficult. As a more tractable alternative, we propose to train a local score network"
>
> with
>
> "Importantly, if the receptive field is $2k + 1$, the network can be trained on segments $x_{i-k:i+k}$ instead of the complete chain $x_{1:L}$, thereby drastically reducing training costs. More generally, we can train a local score network"
>
> to make the benefits less ambiguous.
>
> At inference, the decomposition of the global score into local scores additionally allows for better parallelization, which could be significant depending on available hardware. Concerning the number of corrections, $C$ (see Algorithm 4) has a linear impact on the number of network evaluations at inference, and would have the same impact with a global score network. Hence, doubling the number of corrections roughly doubles the inference time, but not the memory consumption, which stays constant.
>
> > In addition, the paper would benefit from the comparison to more standard data assimilation techniques.
>
> We 100% agree with you. Unfortunately, we were only able to find ad-hoc implementations of 3D-Var and 4D-Var, with barely any documentation. If you point us to a relatively easy-to-use implementation, we'd be happy to oblige. We considered implementing 4D-Var ourselves, but we quickly realized that this would be very hard (which highlights the usefulness of SDA) and beyond the scope of this work.
>
> > As mentioned by the authors, the dynamical model is assumed to be perfectly known or at the very least fixed. [...]
>
> This is correct, but we invite you to consult the clarifications concerning this limitation in the global rebuttal.
>
> **Questions**
>
> > Can the authors highlight the difference/added value between their predictor-corrector sampler and the one in [24] ?
>
> The difference between our predictor-corrector sampler and the one from Song et al. [24] (Algorithms 2 and 3) is the predictor. We employ a predictor called exponential integrator introduced by Zhang et al. [26]. Note that we do not claim our predictor-corrector sampling to be a novel contribution.
>
> > Would it be possible to quantify the closeness to the observations quantitatively [...] ?
>
> Because the observation process $p(y | x_{1:L}) = \mathcal{N}(y | \mathcal{A}(x), \Sigma_y)$ is known, we can measure the likeliness of an observation with respect to an inferred trajectory. However, the likeliness for a single pair $(x_{1:L}, y)$ cannot be interpreted. Instead, as is done for the Lorenz 63 system (see Figure 2), we can compute the expected log-likelihood $E_{q(x_{1:L} | y) p(y)} [ \log p(y | x_{1:L}) ]$, which should be equal to minus the entropy of $\mathcal{N}(0, \Sigma_y)$. In the case of the low and high-frenquency observation processes, the entropies are respectively $\frac{9}{2} \log(2 \pi e 0.05^2) \approx -14.191$ and $\frac{65}{2} \log(2 \pi e 0.25^2) \approx 2.122$, which matches what we observe in Figure 2. It should be noted, however, that this equality is a necessary but insufficient condition for exact posterior inference.
>
> We will add a new section in Appendix computing this expected likelihood for the Kolmogorov system.
>
> > It would be interesting to show several posterior samples on one of these experiments to have an idea of whether the model exhibits a multimodal posterior in this situation [...]
>
> We will add the suggested examples in the Appendix.

---

### Official Review · Reviewer_bwmG · 2023-07-07

**Soundness:** 3 good
**Presentation:** 3 good
**Contribution:** 3 good
**Rating:** 7
**Confidence:** 4

**Summary:**

The authors propose score-based data assimilation framework that relies
on score based generative modeling for trajectory inference/state estimation
of a dynamical model described by an SDE.
To make the procedure efficient, the authors employ three novelties, partly
adopted from existing literature.
They train a score based generative model on short segments of trajectories, and  decompose the posterior score into a prior score and a likelihood score, and approximate the latter with a  Gaussian assuming a Gaussian observation process.
For approximating the prior score, they rely on the Markovian nature of the system and employ local (in time)
prior score approximations.
They present their proposed approach on two non trivial model systems.



I find the contribution interesting, however the novelty the overall ambition
of the project is slightly limited for a conference like NeurIPS. In particular,
I would consider more reasonable the option of trying to identify both
the dynamics and estimate the state of the model, since in most practical
scenarios it is the underlying model dynamics that are unknown. In that respect I find the
use of word inference throughout the text misleading, since the actual problem that
is solved here is not model identification but model estimation and prediction.

Overall I find the paper quite well written, and technically sound. However the novelty and the contribution in the
existing literature is relatively low compared to existing approaches.

**Strengths:**

- the approach does not require simulation through the model equations, that can in principle become too computationally demanding for large systems.

- well written manuscript and nice presentation.

- the proposed method provides an approximation to the entire posterior of the path of the system instead of a point estimate, that in principle allows for further Bayesian inference of the underlying dynamical model.

**Weaknesses:**

- The observation model employed is rather trivial (Gaussian), yet the derivation relies on this assumption. Have you tested the approach with different observation models to test how well this Gaussian approximation holds?

- To my understanding, the trained system is able to generate only
trajectories in equilibrium, and it would fail in non-equilibrium scenarios.
- The authors mention that the introduced approximations introduce errors in the system
but I would expect them to at least quantify the resulting errors
numerically, if theoretical predictions cannot be formulated.

- The authors assume known physical model (both the parametric form and the parameters) that
considerably limits the applicability of the framework for practical applications.

- To my understanding, the noise employed in the numerical experiments is very small, and thereby one could treat the system as deterministic.

**Questions:**

- the system in consideration is assumed to be stochastic, however in Fig.2 C the trajectories seem
pretty deterministic to me. I assume that the individual non-opaque trajectories are individual realisations.
Is the employed noise amplitude 0.025? I consider this negligible noise, especially considering the dynamic range of the system.

- can you quantify either theoretically or empirically the errors incurred by each of the approximations employed in the proposed solution.

- have you tried to predict non-stationary trajectories with the proposed framework? I would think that already the score function estimator
would struggle considerably in this scenario.

- doesn't Eq. 12 only hold in expectation?

**Limitations:**

- the proposed framework assumes an already known dynamical model which is a rather limiting scenario considering both state of the art frameworks and practical applicability.

- according to my understanding, the employed noise in the experiments is weak and the system could be effectively treated as deterministic

- the method is limited only to Gaussian observation model settings.

---

> ### Author Rebuttal · Authors · 2023-08-08
>
> Thank you for your review and reading the manuscript in details. We hope that the following answers will satisfy you.
>
> **Weaknesses & Limitations**
>
> > [...] However, the novelty and the contribution in the existing literature is relatively low compared to existing approaches.
>
> We will address the lack of discussion about the significance of the contributions in Section 5 (Conclusion). We invite you to consult the global rebuttal regarding the novelty of the contributions.
>
> > The authors assume known physical model (both the parametric form and the parameters) that considerably limits the applicability of the framework for practical applications.
>
> We believe this concern is due to some misleading statements in Section 5 (Conclusion), which will be addressed. We invite you to consult the global rebuttal regarding the assumption of a known physical model.
>
> > The observation model employed is rather trivial (Gaussian), yet the derivation relies on this assumption.
>
> Indeed, the observation process is assumed to be Gaussian. However, we argue that this assumption is by no means trivial. It covers all observations that can be expressed as "measurement + Gaussian noise", which corresponds to many real-life scenarios, including some of the most challenging non-linear ones. In fact, an overwhelming amount of literature in the fields of state estimation and Bayesian filtering (Kalman filtering) is based on that Gaussian assumption. In the field of conditional score-based modeling, this is also a very common assumption [19, 24, 34-37] and we are not aware of any literature assuming non-additive measurement noise.
>
> > To my understanding, the trained system is able to generate only trajectories in equilibrium, and it would fail in non-equilibrium scenarios.
>
> In our experiments, we leverage the statistical stationarity of the dynamics to make the local score network index-agnostic, as mentioned at line 153. If stationarity cannot be assumed, for instance in a periodic environment (e.g. seasons), the local score network would simply need to be conditioned with respect to the indices of the states.
>
> However, your comment raises a legitimate point: the score network can only be valid for dynamics it has been trained for, which leads to open questions regarding distribution shifts (e.g. climate change), as mentioned in Section 5.
>
> > The authors mention that the introduced approximations introduce errors in the system but I would expect them to at least quantify the resulting errors numerically, if theoretical predictions cannot be formulated. [...] Can you quantify either theoretically or empirically the errors incurred by each of the approximations employed in the proposed solution?
>
> We already provide quantitative results for Lorenz 63, for which the true posterior is available via the bootstrap particle filter (BPF). We empirically demonstrate that SDA quantitatively converges to the true posterior as $C$ and $k$ increase (see Figure 2). For the Kolmogorov Flow system, the true posterior is intractable with classic posterior inference or data assimilation techniques, and therefore, cannot be quantitatively compared to.
>
> As for theoretical estimation, the impact of the approximations (13) and (15) remains to be quantified, which we leave for future work, as stated in Section 5.
>
> > To my understanding, the noise employed in the numerical experiments is very small, and thereby one could treat the system as deterministic.
>
> Indeed, the noise used for both systems is not large. However, because the systems are chaotic, very small changes in the initial state or small stochasticites can lead to vastly different trajectories. Using more noise would only make the system more chaotic and make inference over long time horizons pointless ($x_L$ would be independent from $x_1$). In fact, increasing the noise levels would make the pseudo Markov approximation (13) more accurate as it would reduce the mutual information between far-away states. To paraphrase, the "low" amount of noise in our experiments is actually a means to challenge our methods and assumptions. Note that, in practice, physical models considered in data assimilation are often deterministic, or present very low amount of noises.
>
> **Questions**
>
> > The system in consideration is assumed to be stochastic, however in Fig. 2C the trajectories seem pretty deterministic to me. [...]
>
> In Figure 2C, trajectories are not independent. They are realizations sampled from the true and estimated posteriors. All trajectories are close to each other because the observation $y$ enables to infer a narrow posterior. Conversely, in Figure 3, the observation (red crosses) leads to a multi-modal posterior, and trajectories diverge even when they start at the same initial states. If we were to represent the prior $p(x_{1:L})$ instead of the posterior $p(x_{1:L} | y)$, one could observe a wide variety of trajectories.
>
> > [...] Is the employed noise amplitude 0.025? I consider this negligible noise, especially considering the dynamic range of the system.
>
> $\Delta = 0.025$ is the variance of the noise at each step. The standard deviation is $0.5$ which is not negligible with respect to the system's domain.
>
> > Have you tried to predict non-stationary trajectories with the proposed framework?
>
> We are actively working on/looking for collaborations with weather and oceanic research centers to apply SDA on operational data, which typically have periodic (daily/yearly) dynamics.
>
> > Doesn't Eq. 12 only hold in expectation?
>
> This equation is valid. We can note that $\nabla_{x_i} \log p(x_{j}) = 0$ and, by definition of a Markov blanket, $p(x_i | x_{\neq i}) = p(x_i | x_{b_i})$.
>
> We believe that this rebuttal addresses most of your concerns and, therefore, kindly ask you to reconsider your score.

---

> > ### Comment · Reviewer_bwmG · 2023-08-21
> >
> > I read the rebuttal to my and the other reviewer’s comments and I updated my score accordingly.
> >
> > I consider the contribution of the paper interesting and the paper well written.
> >
> > My previous concerns regarding the simplicity of the Gaussian observation model are partly unreasonable, if we take into account the rest of this work. However as a comment to the authors there is existing work that identifies latent stochastic dynamics with non-gaussian observation models, i.e. Poisson.
> > Further, the clarification of the authors regarding a known physical model eliminated my concerns of limited contributions of the submitted work.

---

### Official Review · Reviewer_DaY5 · 2023-07-14

**Soundness:** 2 fair
**Presentation:** 3 good
**Contribution:** 2 fair
**Rating:** 6
**Confidence:** 4

**Summary:**

Under a data assimilation setting —where the hidden state of a dynamical system is only accessed through noisy observations— the authors consider the problem of inferring the latent state as well as learning a faithful generative model.  In order to accomplish this, they take a score-based diffusion model approach by placing a diffusion model prior over latent trajectories.  They reveal many practical problems that arise due to the complexity of training the model with (possibly and practically) very high dimensional latent trajectory samples.


**Strengths:**

The paper is well written, and I think the flow and order of presentation was great.  Those who may only be somewhat familiar with denoising diffusion models and latent variables should be able to follow along for the most part.

To make training a diffusion model with the intent to generate high dimensional latent trajectories scale practical, the authors propose pragmatic solutions to problems that arise:

They propose to take advantage of the temporal smoothness of a diffusion model prior by approximating the score function with respect to particular time indices of the perturbed trajectories with the score function of an artificially imposed Markov blanket around that index; they are able to show that in practice, only a few neighbors are needed to create a large enough artificial Markov blanket.

They propose to approximate the conditional likelihood of an observation given a sample from a point in the chain; unlike other proposed methods, they do not have to rescale the residuals

The experiments were interesting, especially, the experiment where the observations are taken to be coarse grained versions of the velocity field; the trajectory sampled from SDA is qualitatively similar to a ground truth sample.  Showing a sample generated from SDA  resembles feeding that sample through the true model was a good demonstration.


**Weaknesses:**

Weaknesses:  It is mentioned in the paper, but by far, the biggest weakness would be that the true underlying dynamics are assumed to be known (or at the very least, can be easily sampled from).  This does slightly bring down the experimental results, in my opinion, since learning the underlying dynamics of an unknown dynamical system is very difficult.

While the paper is well written, and the problem considered is both important and challenging, there is not too much in the way of technical advances; therefore, I think there should have also been more focus on the empirical results.  While I am not certain if many classes of models/methods exist for solving the exact same problem — I am wondering if the authors could have compared their method to other denoising diffusion models that operate on time series e.g. audio or video.


**Questions:**

Can the method only generate trajectories of length L?

Coming from a probabilistic inference and signal processing background, I was wondering if the author considers data assimilation models to be a superset of models that contain state-space models?  Prior to this, I have never encountered this interesting field of data assimilation, and I felt like tying back this model to simpler latent variable models such as those would help increase its scope (mentioned a little at the end, but I think moving this up/fleshing it out could help).


**Limitations:**

As mentioned, the biggest limitation is assumption of the underlying dynamics.

There are not many comparisons against other methods made.

---

> ### Author Rebuttal · Authors · 2023-08-08
>
> Thank you for your review and the legitimate concerns you have raised. We hope that the following answers will satisfy you.
>
> **Weaknesses & Limitations**
>
> > [...] the biggest weakness would be that the true underlying dynamics are assumed to be known (or at the very least, can be easily sampled from).
>
> We believe this concern is due to some misleading statements in Section 5 (Conclusion), which will be addressed. We invite you to consult the global rebuttal regarding the assumption of a known physical model.
>
> > [...] there is not too much in the way of technical advances;
>
> We will address the lack of discussion about the significance of the contributions in Section 5 (Conclusion). We invite you to consult the global rebuttal regarding the novelty of the contributions.
>
> > [...] I think there should have also been more focus on the empirical results.
>
> We respectfully disagree with this statement. Three pages of the manuscript as well as a few pages in the Appendix are dedicated to empirical results, with both quantitative and qualitative assessments. Besides, we already condensed Sections 2 and 3 to the limit and cannot afford to remove any more content.
>
> > [...] I am wondering if the authors could have compared their method to other denoising diffusion models that operate on time series e.g. audio or video.
>
> Indeed, diffusion models have been extensively used for sequence data, including text, audio and video. Some of these methods could have been applied to the first system, Lorenz 63. However, the purpose of Section 4.1 is to compare our approximations against the ground-truth posterior (which is accessible) and study their convergence with respect to hyper-parameters ($k$ and $C$). The comparison with other approximations would have required a lot of space, without added value.
>
> For the second system, Kolmogorov flow, the only contenders would have been video diffusion models (VDMs) [19]. However, as briefly mentioned in Section 5, these models generate sequences autoregressively from "left-to-right" (past-to-future), which prevents from conditioning and sampling all frames simultaneously. Hence, they cannot be used for trajectory posterior inference. We could have built our own vanilla diffusion model for image sequences, but it would have been very expensive to train and irrelevant as a point of comparison.
>
> **Questions**
>
> > Can the method only generate trajectories of length L?
>
> Training requires sequences of length $2k + 1$, but the length $L$ of the generated trajectories is chosen arbitrarily (albeit longer than $2k+1$) at inference.
>
> > I was wondering if the author considers data assimilation models to be a superset of models that contain state-space models?
>
> Indeed, state-space models are a special case of dynamical systems. We chose to frame the paper within the context of data assimilation because of its generality.
>
> We believe that this rebuttal addresses most of your concerns and, therefore, kindly ask you to reconsider your score.

---

> ### Comment · Reviewer_DaY5 · 2023-08-17
>
> The authors have addressed some of my concerns, so I have raised my score to reflect this.

---

### Author Rebuttal · Authors · 2023-08-08

We would like to thank the reviewers for the quality and pertinence of their reviews. All reviewers declare that the methods are sound, interesting and well presented.

The main concerns of reviewers DaY5 and bwmG regard the assumption of a known physical model and the novelty of the contributions. We beleive these concerns are due to some misleading statements in Section 5 (Conclusion) and a lack of discussion about the significance of the contributions, which we propose to address with minor changes in the manuscript. Other comments will be addressed in the per-reviewer rebuttals.

**Assumption of a known physical model**

The introduction starts with the assumption of a known physical model because data assimilation (DA) methods typically and routinely rely on physical models to make up for the lack of data in geophysical systems. In these settings, purely data-driven approaches are simply not applicable. However, SDA is not limited to that assumption and can be applied as long as segments $x_{i-k:i+k}$ ($k \ll L$) are available for training. These segments are not necessarily synthetic and we can apply SDA from data alone. For instance, in cellular biology, the state of some systems can be observed with incredible accuracy, but their dynamics are still relatively unknown.

What we tried to convey at lines 273-280 was that

1. All trajectories are assumed to share the same dynamics, known or unknown.
2. If a physical model is used to generate the training data, one can only expect SDA to be as accurate as the physical model.

We propose to clarify these points in the manuscript by

1. replacing (line 273)

    > Another limitation of our approach is the assumption that the parameters of the physical model are known.

    with

    > Another limitation of our approach is the assumption that the dynamics of the system are shared by all trajectories. In particular, if a parametric physical model is used, all trajectories are assumed to share the same parameters.

2. replacing (line 276)

    > Similarly, SDA relies on the assumption that the physical model is an adequate description of reality. Although we show that it can generalize well to unlikely scenarios (see Figure 5), we can only expect our approach to be as accurate as the physical model itself.`

    with

    > Additionally, if a physical model is used to generate synthetic training data, instead of relying on real data, one can only expect SDA to be as accurate as the model itself.

**Novelty of the contributions**

As noted by reviewers RNbZ and RS1N, our work present several novel contributions, in both the fields of data assimilation and score-based generative modeling. As the significance of these contributions are not explicitly discussed in the manuscript, we propose to add the following sub-section in Section 5.

> **Significance** In addition to its contributions to the field of data assimilation, this work presents new technical contributions to the field of score-based generative modeling.
>
> First, we provide new insights on how to exploit the structure of sets of random variables to build and train score-based generative models. Based on these findings, we are able to generate/infer simultaneously all the states of arbitrarily long Markov chains $x_{1:L}$, while only training score models on short segments $x_{i-k:i+k}$, thereby reducing training costs and the amounts of training data required. The decomposition of the global score into local scores additionally allows for better parallelization at inference, which could be significant depending on available hardware. Importantly, the pseudo-blanket approximation (13) is not limited to Markov chains, but could be applied to any set of variables $x_{1:L}$, as long as some structure is known.
>
> Second, we motivate and introduce a novel approximation (15) for the perturbed likelihood $p(y | x(t))$, when the likelihood $p(y | x)$ is assumed (linear or non-linear) Gaussian. We find that computing the likelihood score $\nabla_{\! x(t)} \log p(y | x(t))$ with this new approximation leads to accurate posterior inference, without the need for stability tricks [37]. This contribution can be trivially adapted to many tasks such as inpainting, deblurring, super-resolution or inverse problems in scientific fields [35-37].

We would also like to emphasize that several papers whose sole purpose were to approximate the likelihood score, were published in major venues [34-37].

---

### Decision · Program_Chairs · 2023-09-21

**Decision:**

Accept (poster)

**Comment:**

All four reviewers argue for accepting this paper. They found it methodologically interesting, well written, and as a possible basis for further work down the stream.

In the rebuttal/discussion, the authors promise to modify the paper (and appendix) with several edits and additional sections. I'm counting on them that the edits will be made as promised. Those edits will help the reader understand the work in better detail and thus ensure a wider impact of the paper.